# Isolation and Characterization of *Escherichia coli* from Brazilian Broilers

**DOI:** 10.3390/microorganisms12071463

**Published:** 2024-07-18

**Authors:** Giulia Von Tönnemann Pilati, Gleidson Biasi Carvalho Salles, Beatriz Pereira Savi, Mariane Dahmer, Eduardo Correa Muniz, Vilmar Benetti Filho, Mariana Alves Elois, Doris Sobral Marques Souza, Gislaine Fongaro

**Affiliations:** 1Laboratory of Applied Virology, Department of Microbiology, Immunology and Parasitology, Federal University of Santa Catarina, Florianópolis 88040-900, Brazil; giuliavpilati@gmail.com (G.V.T.P.); gleidson.salles@zoetis.com (G.B.C.S.); beasavis2@gmail.com (B.P.S.); marianedahmer@gmail.com (M.D.); maariana.eloiss@gmail.com (M.A.E.); doris.sobral@gmail.com (D.S.M.S.); 2Zoetis Industry of Veterinary Products LTDA, São Paulo 04709-111, Brazil; eduardo.muniz@zoetis.com

**Keywords:** avian pathogenic *Escherichia coli*, colibacillosis, Brazilian poultry, antimicrobial resistance, antimicrobial resistance genes

## Abstract

Avian pathogenic *Escherichia coli* (APEC) causes colibacillosis, one of the main diseases leading to economic losses in industrial poultry farming due to high morbidity and mortality and its role in the condemnation of chicken carcasses. This study aimed to isolate and characterize APEC obtained from necropsied chickens on Brazilian poultry farms. Samples from birds already necropsied by routine inspection were collected from 100 batches of broiler chickens from six Brazilian states between August and November 2021. Three femurs were collected per batch, and characteristic *E. coli* colonies were isolated on MacConkey agar and characterized by qualitative PCR for minimal predictive APEC genes, antimicrobial susceptibility testing, and whole genome sequencing to identify species, serogroups, virulence genes, and resistance genes. Phenotypic resistance indices revealed significant resistance to several antibiotics from different antimicrobial classes. The isolates harbored virulence genes linked to APEC pathogenicity, including adhesion, iron acquisition, serum resistance, and toxins. Aminoglycoside resistance genes were detected in 79.36% of isolates, 74.6% had sulfonamide resistance genes, 63.49% showed β-lactam resistance genes, and 49.2% possessed at least one tetracycline resistance gene. This study found a 58% prevalence of avian pathogenic *E. coli* in Brazilian poultry, with strains showing notable antimicrobial resistance to commonly used antibiotics.

## 1. Introduction

In 2023, the global chicken meat production was 102.389 million tons [1]. Brazil is among the largest producers and exporters of chicken meat in the world. According to the Brazilian Association of Animal Protein (ABPA), in 2023, Brazil produced 14.833 million tons of chicken meat, making it the second largest producer in the world. Of this production, 65.35% was destined for the domestic market and 34.650% was destined for exports. Brazil exported 5.139 million tons of chicken meat; hence, it is considered the largest exporter of chicken meat in the world [1].

The bacteria *Escherichia coli* was first described in 1884 by the German microbiologist and pediatrician Theodor Escherich [2]. It has a cosmopolitan distribution, and its various serotypes are intestinal inhabitants found in large numbers in most animals, including humans [3,4,5]. *E. coli* strains can be classified according to their antigenic structure. Thus, there are approximately 180 O, 80 K, and 60 H antigens. The O antigen determines the serogroup; the addition of H antigens and sometimes K antigens determines the serotype [5,6,7].

Colibacillosis is a disease of local or systemic manifestation, usually associated with avian pathogenic *Escherichia coli* (APEC), although not every case of colibacillosis is necessarily caused by APEC [8]. Colibacillosis is one of the most important diseases in poultry production because it leads to significant economic losses due to carcass condemnation and other conditions, such as colisepticemia, peritonitis, pneumonia, pleuropneumonia, airsacculitis, pericarditis, cellulitis, coligranuloma, panophthalmos, omphalitis, oophoritis, osteomyelitis, salpingitis, swollen head syndrome (SCI), and synovitis [5,9,10].

Primary APEC infections are related to management failures, such as high ammonia concentrations, high population density, temperature fluctuations, and other environmental changes that affect birds. When secondary, APEC infections can be associated with respiratory viruses such as Newcastle disease, Infectious Bronchitis Virus, and avian Metapneumovirus or with bacteria such as *Mycoplasma gallisepticum* and *Mycoplasma synoviae* [5,8,9].

Different virulence and pathogenicity factors are used by APEC strains to cause colibacillosis in birds. APEC strains utilize different virulence and pathogenicity factors to establish an infection and cause disease in birds. The main virulence factors include proteins that facilitate adhesion and invasion, elements involved in toxin production, secretion systems, and mechanisms for antibiotic resistance [5,11]. A set of five genes have been identified and considered the minimal virulence predictors for an isolate to cause colibacillosis, distinguishing APEC from avian fecal *Escherichia coli* (AFEC). These genes are *hlyF* (putative avian hemolysin), *iutA* (aerobactin siderophore receptor gene), *iss* (episomal enhanced serum survival gene), *iroN* (salmochelin siderophore receptor gene), and *ompT* (outer membrane protease gene) [3,10,12].

Several antimicrobial agents from different classes are used for the treatment of colibacillosis, including β-lactams (penicillins, cephalosporins), aminoglycosides, lincosamides, tetracyclines, sulfonamides, quinolones, and fluoroquinolones [5,13,14]. Currently, many of the antimicrobials used in poultry production are also utilized in human medicine, a fact that raises concerns about the potential transfer of antimicrobial resistance genes between animals and humans [15].

Antimicrobial agents can be natural, semisynthetic, or synthetic substances, which act by inhibiting or killing microorganisms and are used in the treatment of different types of infections caused by bacteria, viruses, fungi, and parasites [16]. Furthermore, antibiotics have been widely used as growth promoters and as metaphylactic agents in animal production. Such practices, however, increase selective pressure and may favor the development of resistance to these substances [14,17,18,19].

Antimicrobial resistance can be naturally developed or acquired [20,21]. Resistance can arise through mutation processes, in which genes, normally present in the bacterial genome, mutate to a form that renders the antibiotic ineffective. Genes encoding antimicrobial resistance can be transferred between bacteria through classical mechanisms of horizontal transfer, such as conjugation, transformation, and transduction, as well as other pathways, like membrane vesicles [14,20,21,22,23,24].

The emergence of resistant and multiresistant bacteria to antibiotics is a growing and worldwide health problem, encouraging the search for alternatives that replace the use of antibiotics. Disinfectants are a good alternative for controlling the growth of microorganisms, and they are used in the food industry, in the agricultural industry, in health establishments, in homes, and in pharmaceutical products [25,26]. Different disinfectants such as formaldehyde, quaternary ammonium compounds (QACs), hydrogen peroxide, and sodium hypochlorite are commonly used on poultry farms [27,28].

Given the importance and role of Brazilian poultry farming, the economic impacts caused by APEC infection, and the challenges of antimicrobial resistance, the characterization of APEC strains is important to understand the pathogenesis of colibacillosis and to develop effective prevention and control strategies. Therefore, this study aims to determine the prevalence of *Escherichia coli* in broiler chickens from six Brazilian states and characterize the isolates obtained from the femurs of necropsied birds. 

## 2. Materials and Methods

### 2.1. Sample Collection

Femur samples were collected between August and November 2021 from chicken carcasses (*Gallus gallus domesticus*) necropsied in the field for sanitary inspection and provided for the present study. The necropsied birds were between 13 and 32 days old. A total of 100 batches were evaluated in 6 different Brazilian states. Samples came from the states of Paraná (*n* = 30), Santa Catarina (*n* = 15), Rio Grande do Sul (*n* = 15), São Paulo (*n* = 10), Minas Gerais (*n* = 10), and Ceará (*n* = 20), which represent approximately 80% of the chicken meat production in Brazil [1].

Anamnesis and clinical evaluation of the batches were performed, and batches were selected based on a history of respiratory problems, as well as clinical manifestations associated with the respiratory tract, such as sneezing, rattling, and nasal discharge. 

For each batch, three femurs were collected for *E. coli* research. The femurs were chosen for the isolation of *E. coli* because they were collected intact, which reduced the chances of bone marrow contamination. Additionally, there was a high probability of being highly pathogenic, as they were circulating in the bone marrow. The samples were received and processed at the Laboratory of Applied Virology (LVA) of the Federal University of Santa Catarina (UFSC). 

All biological samples assessed in this study were donated by farms subject to regular inspection routines, eliminating the need for approval from an ethics board as they were residual samples collected in routine of health surveillance services—Consultation with the Ethics Committee on the Use of Animals (no 4434190521/Federal University of Santa Catarina).

### 2.2. Escherichia coli Isolation 

For *E. coli* isolation, femurs were processed aseptically, and the bone marrow was collected with a swab, which was suspended in saline buffer. The swabs were inoculated in MacConkey agar and incubated at 37 °C for 24 h. Typical *E. coli* colonies (pink, precipitated colonies) were isolated. Isolates were preserved in a repository and biobank in Luria–Bertani (LB) broth (KASVI, Marbella, Spain) with glycerol (5:1), and maintained at −80 °C for subsequent genomic DNA extraction.

### 2.3. APEC Molecular Confirmation

The femur isolates were subjected to qualitative polymerase chain reaction (PCR) to identify APEC strains. Genomic DNA extraction was performed using an adapted phenol–chloroform method [29]. 

Cryopreserved colonies were grown in Luria–Bertani (LB) broth for 6 h at 37 °C, then cultured on MacConkey agar, and incubated for 24 h at 37 °C. A colony was selected and inoculated again in 25 mL of LB broth for 18 h at 37 °C. After incubation, the medium was centrifuged (700× *g* for 10 min), the supernatant was discarded, and the pellet was washed twice with PBS 1× pH 7.2. The sample was centrifuged again, and the pellet was separated into a microtube. For sample lysis, the pellet was suspended in 200 μL of lysis buffer (10 mM Tris HCl pH 7.4, 10 mM NaCl, 25 mM ethylenediaminetetraacetic acid (EDTA) pH 8, 1% sodium dodecyl sulfate (SDS)), containing 26.2 μL of Proteinase K, and it was incubated in a thermoblock at 56 °C for 30 min. For DNA extraction, an equal volume of equilibrated phenol was added, homogenized by inversion, and centrifuged for 10 min at 14,000 ×*g* at room temperature. The aqueous phase was transferred to a new microtube, and the same volume of a phenol–chloroform solution (1:1) was added, homogenized by inversion, and centrifuged for 10 min at 14,000× *g* at 4 °C. The aqueous phase was transferred to a new microtube, and an equal volume of chloroform was added, homogenized by inversion, and centrifuged for 10 min at 14,000× *g* at 4 °C. The aqueous phase was transferred to a new microtube, and to each tube, 2.5 times the total volume of ice-cold 100% ethanol was added. The microtubes were kept at −20 °C for 1 h. Tubes were centrifuged for 30 min at 14,000× *g* at 4 °C. The supernatant was discarded, and the pellet was washed twice with 500 μL of ice-cold 70% ethanol and centrifuged for 10 min at 14,000× *g* at 4 °C. The pellet was dried in a thermoblock at 37 °C. Finally, the pellet was suspended in 50 μL of DNase- and RNase-free water (QIAGEN, Inc., Valencia, Spain, EUA). DNA was quantified by optical density using NanoVue™ spectrophotometry and stored at −20 °C.

For the qualitative PCR reactions, the genes (*iroN*, *ompT*, *hlyF*, *iss*, and *iutA*) were used as minimum virulence predictors of APEC (Table 1). The reagents for amplifying the five gene targets were used at the following concentrations: 2 mM magnesium chloride, 0.25 mM deoxyribonucleotide phosphates (dNTPs), 0.3 μM of each primer, 1 U of GoTaq^®^ DNA Polymerase (Promega, Madison, WI, USA), 1× Green GoTaq^®^ Reaction Buffer (Promega, Madison, WI, USA), 3 μL of sample, and sterile ultrapure water to complete 25 μL. The reactions were performed in a thermocycler (TECHNE Flexigene, Burlington, VT, USA), using the following cycling parameters: 94 °C for 2 min; 35 cycles of 94 °C for 30 s, 63 °C for 30 s, 68 °C for 3 min; and a final cycle of 72 °C for 10 min [12].

The samples were subjected to horizontal agarose gel electrophoresis at 1%, using GelRed as a DNA intercalating agent. The sizes of the amplicons were determined by comparison with a low-molecular-weight (LMW) marker.

### 2.4. Antimicrobial Sensitivity Testing

The disk diffusion method was used to perform the antibiotic sensitivity test, following the Kirby–Bauer (KB) methodology [31]. Eleven antibiotics (Laborclin) were tested: Nalidixic Acid (30 μg), Ampicillin (10 μg), Azithromycin (15 μg), Ceftiofur (30 μg), Ceftriaxone (30 μg), Enrofloxacin (5 μg), Gentamicin (10 μg), Lincomycin/Spectinomycin (109 µg), Nitrofurantoin (300 µg), Norfloxacin (10 µg), and Sulfamethoxazole/Trimethoprim (25 µg). Isolated strains were inoculated in tubes containing LB broth and incubated at 37 °C for 18 h. After incubation, diluted cultures were swabbed onto plates containing Mueller–Hinton agar (HIMEDIA, Mumbai, India). Antimicrobial disks were added, and the plates were incubated for 18 h at 37 °C. The plates were read by measuring the diameter of the inhibition zones using a vernier. A resistance pattern of the inhibition zone was considered according to the VET01S Performance Standards for Antimicrobial Disk and Dilution Susceptibility Tests for Bacteria Isolated from Animals, 6th Edition, and M100 Performance Standards for Antimicrobial Susceptibility Testing, 33rd Edition, established by the Clinical and Laboratory Standards Institute (CLSI) [32,33].

### 2.5. Whole Genome Sequencing

For sequencing, genomic DNA libraries were prepared using an Illumina DNA Prep—Nextera kit (Illumina, Inc., San Diego, CA, USA) and quantified with a Collibri Library Quantification Kit (Invitrogen Inc., Carlsbad, CA, USA) following the manufacturer’s recommendations. A Nextseq system (Illumina) was employed to generate raw reads based on 300 cycles, in a paired-end sequencing configuration (2 × 150 bp reads).

Raw data obtained from the MiSeq platform (Illumina) were processed using the Phred quality score. Reads with a Q score below 20 were excluded from the analyses, and adapters or segments with poor quality were also discarded.

After quality control (QC), genome assembly was performed using the company’s proprietary pipeline, oneshotWGS v1.9. OneshotWGS v1.9 integrates a set of bioinformatics tools commonly used by the scientific community. One of these programs is the A5 assembly, which includes additional steps for adapter trimming, quality filtering, and error correction to generate scaffolds [34]. Chimeric segments were removed at the end of the assembly steps (best assembly). Assembly statistics (total scaffolds, GC content percentage, N50, L50, etc.) were determined using QUAST 5.2.0 software [35].

Sequencing and genome assembly data were deposited in the National Center for Biotechnology Information (NCBI) database under Bioproject accession number PRJNA917297.

Genome annotation was performed using g Prokka 1.14.6 software [36]. For this step, custom proprietary (Neoprospecta, Florianópolis, Brasil) databases were used, consisting of curated gene sequences obtained from public databases such as Pfam, GenBank, nt/nt, etc. A similar protocol was applied for virulence and resistance genes.

### 2.6. In Silico Analysis

#### 2.6.1. Species Confirmation

In silico confirmation of the species was conducted, and the best assembly was used for the species identification process, implemented by the company’s pipeline module, neogSpecies. This module, written in Python 3.11.9, applied an Average Nucleotide Identity (ANI) analysis to estimate genome species (cutoff: 97%). ANI is the standard method for defining a prokaryotic species [37].

#### 2.6.2. Serogroup Identification

Serogroup determination utilized all genome sequencing data from the sample, and the Sorotypefinder 2.0 program (https://www.ncbi.nlm.nih.gov/pmc/articles/PMC4508402/ (accessed on 20 May 2023)) was employed for serogroup prediction.

#### 2.6.3. Virulence, Antimicrobial, and Disinfectant Resistance Gene Detection

The prediction of virulence genes was carried out using the virulencefinder 2.0 program (https://pubmed.ncbi.nlm.nih.gov/24574290/ (accessed on 20 May 2023)) with a minimum identity of 90%, based on the data obtained from whole genome sequencing.

The presence of antimicrobial resistance genes was assessed with complete genome sequencing data, using the Abricate 1.0.1 program (https://github.com/tseemann/abricate (accessed 24 May 2023)), with the Resfinder database version (accessed 24 May 2023), applying a minimum coverage and identity of 80% [38].

## 3. Results

### 3.1. Batch History

From the batches analyzed, 61% had a history of respiratory problems, septicemia, and/or mortality, with unknown etiology. During sample collection, 42% of the batches exhibited clinical signs of respiratory symptoms, such as rales, sneezing, nasal discharge and discharge, infraorbital sinus enlargement, and swollen head.

Among the batches from the Southern states (Paraná, Santa Catarina, and Rio Grande do Sul), 13.3% used antibiotics during the bird housing phase. The drugs used were ciprofloxacin, sulfachlorpyridazine+trimethoprim, and florfenicol. In the Southeast Region (Minas Gerais and São Paulo), only one batch (5%) showed clinical signs and was treated with ciprofloxacin on the day of collection.

### 3.2. Escherichia coli Isolation and APEC Confirmation

A total of 63 isolates characteristic of *E. coli* were obtained from the femurs, and these isolates were subjected to whole genome sequencing. Through sequencing, all 63 isolates (100%) were identified as *Escherichia coli*.

Using qualitative PCR, out of the 63 *E. coli* isolates, 58 (92%) exhibited between three and five of the genes considered minimum virulence predictors for APEC strains, thus confirming their classification as APEC. Among the 63 analyzed isolates, 40 (63.4%) exhibited all five genes, 14 (22.2%) had four genes, 4 (6.3%) had three genes, 4 (6.3%) had one to two genes, and in one batch, none of the five genes were detected (Table 2). Regarding APEC isolates, 96.5% carried the *ompT* and *iss* genes, 93.1% had *hlyF*, 94.8% contained the *iutA* gene, and 89.6% harbored *iroN*.

### 3.3. Phenotypic Antimicrobial Resistance

The obtained diameters were compared with the cutoff points established by the CLSI [32,33].

The overall resistance rates found were 66.67% for Ampicillin, 7.94% for Azithromycin, 44.44% for Ceftriaxone, 44.44% for Ceftiofur, 39.68% for Enrofloxacin, 30.16% for Gentamicin, 19.95% for Lincomycin/Spectinomycin, 69.84% for Nalidixic Acid, 7.94% for Nitrofurantoin, 14.29% for Norfloxacin, and 42.86% for Sulfamethoxazole/Trimethoprim.

In the Southern Region, the resistance rates found for *E. coli* isolates were 64.71% for Ampicillin, 2.94% for Azithromycin, 47.06% for Ceftriaxone, 47.06% for Ceftiofur, 32.35% for Enrofloxacin, 23.53% for Gentamicin, 14.71% for Lincomycin/Spectinomycin, 61.76% for Nalidixic Acid, 2.94% for Nitrofurantoin, 8.82% for Norfloxacin, and 26.47% for Sulfazotrim.

In the Southeast Region, the results found for *E. coli* isolates indicate a resistance to Ampicillin of 57.89%, 47.37% for Ceftriaxone, 47.37% for Ceftiofur, 44.11% for Enrofloxacin, 26.32% for Gentamicin, 31.58% for Lincomycin/Spectinomycin, 68.42% for Nalidixic Acid, 5.26% for Nitrofurantoin, 5.26% for Norfloxacin, and 63.16% for Sulfazotrim. All isolates were sensitive to Azithromycin.

In the Northeast Region, the resistance indices found that the *E. coli* isolates indicate a resistance to Ampicillin of 70%, 30% for Azithromycin, 30% for Ceftiofur, 30% for Ceftriaxone, 60% for Enrofloxacin, 40% for Gentamicin, 10% for Lincomycin/Spectinomycin, 80% for Nalidixic Acid, 20% for Nitrofurantoin, 40% for Norfloxacin, and 40% for Sulfazotrim.

Figure 1 represents the general resistance rates and the rates for each region.

### 3.4. Serogroups

Whole genome sequencing was used to determine *E. coli* serogroups. It was possible to identify the serotype of a total of 60 isolates (92%), distributed among 40 serogroups (Figure 2). The predominant serogroups were O128 and O53, both with a frequency of 6.8%. Serogroups O78 and O16 occurred with a frequency of 5.1%, while O2, O25, O5, O110, O71, and O109 occurred with a frequency of 3.4%. The remaining serogroups appeared at a lower frequency (1.7%).

### 3.5. Detection and Distribution of Genes Associated with Virulence

The sequenced isolates carried between 4 and 27 virulence genes, and the prevalence of these genes ranged from 1.58% to 100%. Of the sequenced isolates, 88% harbored at least one gene related to adhesion (*eae*, *papA_F11*, *papA_F19*, *papA_F20*, *papC*, *hra*, *iha*, *lpfA*, and *tsh*), while 93.65% harbored at least one gene related to iron acquisition systems (*chuA*, *fyuA*, *ireA*, *irp2*, *iroN*, *iucC*, *iutA*, *sitA)*. Regarding serum resistance factors, 96.82% of the isolates harbored at least one virulence gene (*iss*, *kpsE, kpsMII*, *kpsMIII_K98*, *kpsMII_K1*, *kpsMII_K5*, *neuC*, *ompT*, *traT*). All isolates (100%) carried at least one gene encoding toxins (*astA*, *cma*, *cvaC*, *hlyE*, *hlyF*, *usp*, *vat*). Table 3 covers the virulence genes found and their prevalence.

The *terC* (Tellurium ion resistance protein) and *gad* (Glutamate decarboxylase) genes were detected in 100% of the isolates.

Genes encoding adhesins are related to processes such as adhesion, motility, biofilm formation, and survival in macrophages. Of the isolates, 69.84% harbored the *hra* gene, 60.31% harbored the *lpfA* gene, 23.8% harbored the *pap* gene, and 34.92% harbored the *tsh* gene.

The ability to resist serum is one of the factors related to APEC strains; 96.82% of the isolates harbored genes related to serum resistance, of which 82.53% harbored the *traT* gene.

Genes related to iron acquisition were detected in 59 isolates, with the most prevalent being *chuA* (41.26%), *fyuA* (31.74%), *iucC* (53.96%), irp2 (31.74%), and *sitA* (61.90).

The *cvaC* gene was detected in 30.15% of the isolates and the *hlyE* gene was detected in 77.77% of the samples.

### 3.6. Detection and Distribution of Antimicrobial and Disinfectant Resistance Genes

Out of the analyzed isolates, all of them showed at least one antimicrobial resistance gene (ARG), and all of them (100%) contained the *formA* gene, a formaldehyde resistant gene. The resistance genes detected in 20% or more of the isolates were *sul2* (60.31%), *sitABCD* (57.14%), *sul1* (52.38%), *ant(3*″*)-Ia* (50.79%), *qacE* (50.79%), *aac(3)-VIa* (36.5%), *aph(6)-Id* (31.74%), *tet(A)* (32.74%), *tet(B)* (20.63%), *aadA2* (20.63%), and *aph( 3*″*)-Ib* (20.63%).

In this study, 78.1% of the APEC isolates harbored at least one of the aminoglycoside resistance genes (*aac(3)-IId*, *aac(3)-IVa*, *aac(3)-VIa*, *aadA12*, *aadA2*, *ant(2*″*)-Ia*, *ant(3*″*)-Ia*, *aph(3*′*)-Ia*, *aph(3*″*)-Ib*, *aph(4)-Ia*, and *aph(6)-Idii*). At least one of the predicted sulfonamide resistance genes (*sul1*, *sul2*, and *sul3*) was found in 73.4% of the isolates.

Similarly, at least one predicted β-lactam resistance gene was found in 64.06% of the isolates, including *blaCMY-2*, *blaCTX-M-1*, *blaCTX-M-15*, *blaCTX-M-164*, *blaCTX-M-2*, *blaCTX-M-55*, *blaCTX-M-8*, *blaSHV-12*, *blaTEM-10*6, *blaTEM-141*, *blaTEM-1A*, and *blaTEM-1B*.

For tetracycline resistance genes, *tet(A)*, *tet(B)*, and *tet(D)*, 48% of the isolates harbored at least one tetracycline resistance gene.

Table 4 covers the antimicrobial classes, the antimicrobial resistance genes found, and their prevalence.

Among the resistance genes detected, those present in more than 20% of the samples from the Southern Region were *aac(3)-VIa* (35.29%), *aadA2* (26.47%), *ant(3*″*)-Ia* (50%), *aph(6)-Id* (29.41%), *blaCTX-M-2* (20.58%), *sul1* (47.05%), *sul2* (47.05%), and *tet(A)* (29.41%).

In the Southeast Region, the most prevalent genes among the batches were *aac(3)-Vla* (36.84%), *ant(3*″*)-Ia* (52.63%), *aph(3*″*)-Ib* (21.05%), *aph(6)-Id* (31.57%), *sul1* (63.15%), *sul2* (73.68%), *tet(A)* (31.57%), and *tet(B)* (26.31%).

In the Northeast Region, the resistance genes detected in more than 20% of the samples were *aac(3)-Via* (40%), *aadA2* (20%), *ant(3*″*)-Ia* (50%), *aph(3*″*)-Ib* (40%), *aph(3*′*)-Ia* (20%), *aph(6)-Id* (40%), *blaTEM-1* (20%), *blaTEM-1B* (20%), *dfrA12* (20%), *floR* (40%), *qnrB19_1* (20%), *sul1* (50%), *sul2* (80%), *tet(A)* (40%), and *tet(B)* (30%).

Regarding resistance genes associated with disinfectants, 100% of the strains harbored a formaldehyde resistance gene (*formA*), 50.79% of the APEC strains harbored the *qacE* gene, and 57.14% of the isolates harbored the *sitABCD* gene.

## 4. Discussion

*E. coli* was isolated from the femurs of chickens from different regions of Brazil. Qualitative analysis by PCR showed that 92% of the isolates tested were characterized as APECs based on the presence of at least three of the five minimal predictors of virulence [12]. In comparison, a study in Poland demonstrated that 43% (124/290) of all tested strains were characterized as APEC; in northern Egypt, 51.85% (28/54) of the batches were positive for APEC, while in Nepal, 90% (45/50) of colibacillosis isolates were identified as APECs [39,40,41].

The percentage of genes in the APEC isolates was 96.5% for *ompT,* 96.5% for *iss,* 93.1% for *hlyF,* 94.8% for *iutA,* and 89.6% for *iroN.* Other studies conducted in Brazil, involving chickens and turkeys, reported similar values. In these studies, the prevalence of APEC in the isolates was 58.6% and 84.34%, with a gene frequency of 98.8% for *iroN*, 96.3% for *iss*, 81.5% for *iutA*, and 100% for *hlyF* and *ompT* [42,43]. A study conducted in Egypt found a gene prevalence of 93.3% for *iss* and 46.6% for *iutA* [40].

Similar to other pathogenic *Escherichia coli* strains, avian pathogenic *E. coli* (APEC) harbor a wide variety of virulence genes that distinguish them from commensal strains. Among these, the ten genes most frequently associated with APEC strains are *iss*, *tsh, iroN*, *ompT*, *iutA*, *cvaC*, *hlyF*, *iucD*, *papG allel* (*II/III*), and *papC* [44]. These genes are located on chromosomes or plasmids, such as the ColV and ColBM plasmids. Virulence factors commonly associated with APEC strains include adhesins, toxins, iron acquisition mechanisms, invasins, and plasmids [5,45,46].

In this study, all isolates (100%) harbored the resistance genes *tetC* (Tellurium ion resistance protein) and *gad* (Glutamate decarboxylase). Tellurite, a Tellurium oxyanion, is toxic to bacteria due to its oxidative capability. However, *Escherichia coli* shows high resistance to tellurite, and among the genes associated with this resistance are *terB*, *terC*, *terF*, *terX,* and *terY3* [47]. Previous studies have described the presence of the *terC* gene in all *E. coli* isolates analyzed [48,49].

The enzyme glutamate decarboxylase (*gad*) transforms glutamate (Glu) and a proton into gamma-aminobutyric acid (GABA) and carbon dioxide, with pyridoxal 5′-phosphate (PLP) acting as a cofactor [50,51]. The gad system is a common mechanism found in bacteria that allows them to survive and adapt to acidic environments [51]. A previous study described the presence of the *gad* gene in 100% of *E. coli* isolates tested [48].

In addition to the five genes used as minimum predictors, various studies describe other genes associated with APEC, including *iucD*, *hlyE*, *irp2*, *papC*, *cva/cvi*, and *tsh* [5,44,45]. In this study, the *pap* gene was detected in 23.80% of the isolates, and previous studies described its presence in 15% of systemic isolates and 26% of cellulitis isolates [52]. The *trat* gene was detected in 82.53% of the isolates tested in this study. Similarly, previous studies detected the *traT* gene in 82% of systemic isolates and in 62.29% of colibacillosis isolates [46,52].

It is essential to note that the use of the five virulence genes alone is insufficient to determine the virulence potential of strains. The correlation between the presence of two plasmid markers, APEC *hlyF* and *ompT*, which are among the most conserved plasmids, with multilocus sequence typing (MLST) or serogrouping can be used to identify highly virulent APEC strains [8]. In the present study, the isolates were obtained from the bone marrow of femurs. To reach the femur, the agent must have the ability to enter the bloodstream, which indicates the potential of the bacteria to be highly pathogenic. Additionally, PCR screening with the five minimum predictors was used, along with the identification of the serogroups of these isolates [12].

Different O serogroups have been associated with colibacillosis; however, the most common ones linked to these cases, found in various studies worldwide, are O78, O2, O1, O18, O35, O36, O109, O115, and O111 [5,45]. In this study, the combined prevalence of these most prevalent serogroups represents a total of 18.7% of the classified isolates in this study.

Serogroups O128 and O53 are not commonly associated with cases of colibacillosis; however, they were the most prevalent among the isolates analyzed in this study. Previous studies characterized isolates from broilers with a history of respiratory symptoms, pericarditis, perihepatitis, and airsacculitis as belonging to serogroup O128 [53,54]. In another study that used samples from birds with lesions characteristic of colibacillosis, 8.9% of the isolates evaluated belonged to serogroup O53 [55]. These data demonstrate the diversity of serogroups among APEC strains. Among the factors that can influence the geographical variation in these serogroups, the genetic diversity of bacterial strains and exposure to various sources of infection are highlighted [5,45].

The resistance of *Escherichia coli* to different classes of antimicrobials has been described. APEC is often resistant to tetracyclines, sulfonamides, ampicillin, and streptomycin. However, this resistance profile varies according to the geographic location and bird characteristics [5,11].

For this study, the antibiotics most widely used in poultry farming were selected, with some of them also employed in human treatment, such as ceftriaxone, nitrofurantoin, and sulfamethoxazole/trimethoprim. All (100%) *E. coli* isolates were resistant to at least one antibiotic, displaying diverse antibiotic resistance profiles. Sixty percent of the strains exhibited resistance to more than three classes of antibiotics, categorizing them as multidrug-resistant. In Brazil, multidrug resistance rates of 54.6%, 78.9%, 94.2%, and 71% have been observed in APEC isolated from cases of airsacculitis, cellulitis, commercial poultry, and chicken carcasses, respectively [56,57,58,59]. Studies conducted in China, Poland, and Thailand reported multidrug resistance rates of 100%, 81.1%, and 80%, respectively [60,61,62].

In this study, the most prevalent resistance genes among the isolates belong to the classes of aminoglycosides, sulfonamides, β-lactams, and tetracyclines. In veterinary medicine, aminoglycosides are widely used to treat bacterial infections in various animal species, including birds. These antibiotics are effective against a variety of Gram-negative bacteria, such as *Escherichia coli*, *Salmonella*, and *Pseudomonas aeruginosa* [63]. Gentamicin is one of the most commonly used aminoglycosides in veterinary medicine. A high presence of aminoglycoside resistance genes was reported in this study, similar to the high rates described previously in Pakistan [64].

Resistance to sulfonamides can be explained by their extensive use in the treatment of infections caused by Gram-negative bacilli and their wide availability on the market [65]. Previous studies described prevalences of sulfonamide resistance genes in 70% and 89.3% of isolates, rates close to those described in this work [14,64]. Another study described the sulfonamide resistance genes, *sul1* and *sul2*, in 6.3% and 25.3% of isolates, respectively [62].

Due to their wide availability and affordable cost, tetracyclines are a class of antibiotics commonly used in veterinary medicine to treat a variety of bacterial infections in animals. They are effective against a wide range of bacteria, including both Gram-positive and Gram-negative bacteria, and are commonly used to treat respiratory, urinary tract, and skin infections in animals [65]. In Jordan, 90.7% of APEC isolates harbored at least one tetracycline resistance gene (*tet(A)* and *tet(B)*) [14], while another study demonstrated that 16.4% of isolates from cases of colibacillosis harbored tetracycline resistance genes (*tet(A)*) [46].

Beta-lactams are widely used in veterinary medicine. One of the main factors involved in high rates of resistance to beta-lactams is the production of extended-spectrum beta-lactamases (ESBLs) by Enterobacteriaceae species [66,67]. This study indicated a prevalence of 64.06% of isolates harboring at least one beta-lactam resistance gene. Here, the presence of the *blaTEM* gene was found in 43.3% of isolates. Similarly, studies conducted in Thailand and Jordan described its presence in 72.9% and 43.3% of isolates, respectively [14,62].

Constant exposure to low concentrations of disinfectant residues can result in increased bacterial tolerance. This increased tolerance may lead to greater bacterial adaptive resistance to antibiotics and the disinfectants themselves, enhancing the ability of bacteria to survive various environmental stresses [68,69]. Studies suggest a relationship between the use of disinfectants and the transfer of antimicrobial resistance genes [70,71,72]. In this study, three genes associated with resistance to disinfectants used in the production chain were found, namely formaldehyde, quaternary ammonium compounds (QACs), and hydrogen peroxide.

A study described that certain strains of *E. coli* have a formaldehyde resistance mechanism that involves enzymatic degradation of the compound by a variant of the enzyme present on a plasmid [73]. A study conducted in Germany described how phenotypic resistance to formaldehyde was associated with the presence of this gene [74]. In this study, all analyzed strains contained a formaldehyde dehydrogenase gene (*formA*).

QACs are widely used in the poultry industry due to their relatively low toxicity, good antibacterial properties, and non-irritating, non-corrosive, and reasonably effective properties in the presence of organic matter [75]. The *qacE* gene is widely spread in Gram-negative bacteria, mainly in Enterobacteriaceae. In this study, 50.79% of APEC strains contained the *qacE* gene. The presence of the *qacEΔ1* gene, described as a *qacE* deletion mutation, has been reported in some studies and appears with a relatively high frequency in isolates [75,76,77].

The SitABCD operon, initially described in *Salmonella enterica*, consists of four distinct regions. The *sitA* gene encodes a periplasmic binding protein that captures manganese and iron. The *sitB* gene encodes the ATP-binding component, providing energy for ion transport across the cell membrane. The *sitC* gene encodes a permease that facilitates the active transport of ions across the membrane. Finally, the *sitD* gene encodes the inner-membrane component of the system, aiding in the transportation and incorporation of ions into the bacterial cytoplasm. Collectively, these genes form an ABC transport system that imparts resistance to the bactericidal effects of hydrogen peroxide and plays a crucial role in the uptake and homeostasis of manganese and iron in bacterial cells [78]. In the current study, 57.14% of the isolates harbored the *sitABCD* gene. In cases of avian mortality associated with colibacillosis, it was observed that the gene *sitABCD* was present in 71.3% of the bacterial isolates [79]. This suggests a potential role of the SitABCD operon in the survival and adaptation of *Escherichia coli* under conditions associated with colibacillosis in poultry.

## 5. Conclusions

This study identified the prevalence of avian pathogenic *Escherichia coli* (APEC) in 58% of the analyzed broiler chicken batches, encompassing commercial poultry-producing regions in Brazil. Additionally, it demonstrated a broad diversity of serogroups distributed throughout the country. These findings underscore the widespread occurrence of pathogenic *E. coli* strains in broiler chicken populations, emphasizing the need for continued monitoring and research to better understand and manage these bacterial infections in the poultry industry.

In this study, all isolates examined harbored at least one virulence gene associated with the pathogenicity of APEC. The majority of the isolates carried genes related to adhesion, iron acquisition systems, and serum resistance factors. Additionally, all isolates demonstrated the presence of genes associated with toxin production.

The characterization of the antimicrobial resistance profile revealed a significant presence of multidrug-resistant strains, not only to antibiotics commonly used in animal production but also to antibiotics frequently employed in human infection treatment.

In the broader context, antimicrobial resistance is recognized as a global issue tied to the concept of One Health, which integrates environmental, human health, and animal health considerations. Integrated strategies, grounded in the One Health approach and addressing all three domains, emerge as potential solutions to combat antimicrobial resistance.

These results highlight the importance of monitoring APEC serotypes and resistances through pharmacovigilance. Surveillance plays a critical role in both evaluating the effectiveness of interventions against this challenge and investigating events, ultimately aiming at the identification and prevention of antimicrobial resistance.

## Figures and Tables

**Figure 1 microorganisms-12-01463-f001:**
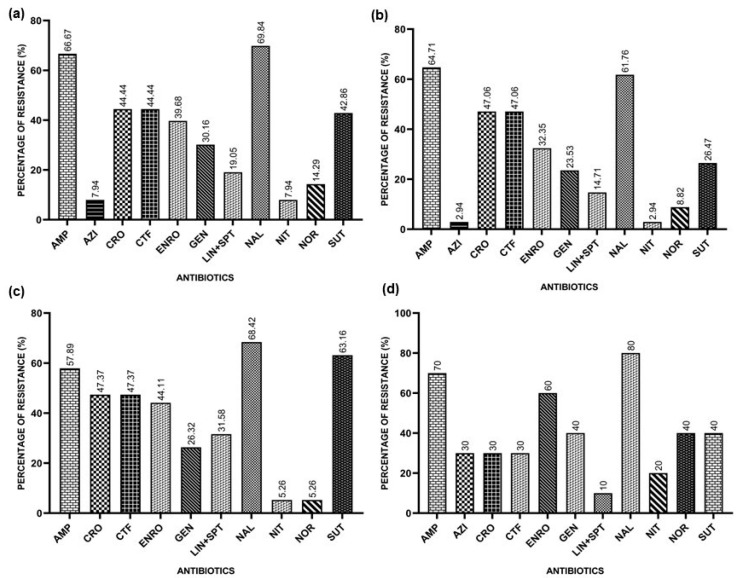
Resistances presented by APEC isolates against the tested antimicrobials: Ampicillin (AMP), Azithromycin (AZI), Ceftiofur (CFT), Ceftriaxone (CRO), Enrofloxacin (ENRO), Gentamicin (GEN), Lincomycin/Spectinomycin (LIN+SPT), Nalidixic Acid (NAL), Nitrofurantoin (NIT), Norfloxacin (NOR), and Sulfazotrim (SUT). (**a**) Resistances presented by APEC isolates against the tested antimicrobials in Brazil; (**b**) resistances presented by the APEC isolates from the Southern Region; (**c**) resistances presented by APEC isolates from the Southeast Region; (**d**) resistances presented by APEC isolates from the Northeast Region.

**Figure 2 microorganisms-12-01463-f002:**
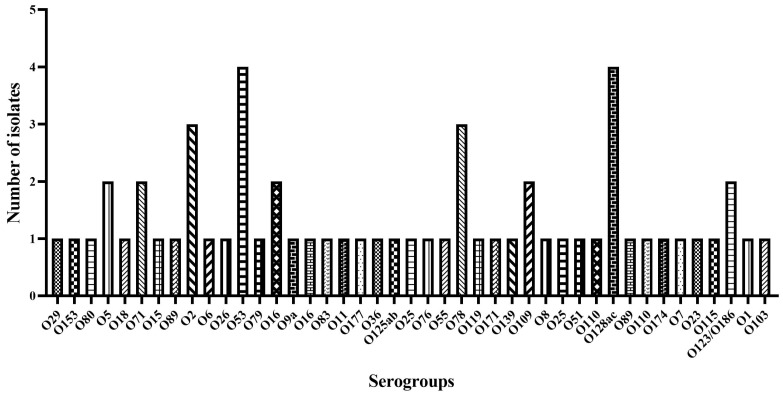
*E. coli* serogroup distribution using the program Sorotypefinder 2.0.

**Table 1 microorganisms-12-01463-t001:** Target gene, sequence of primers, amplicon size, and reference.

Target Gene	Primer Sequence	Amplicon Size	Reference
*iroN*	5′-AAGTCAAAGCAGGGGTTGCCCG-3′ 5′-GATCGCCGACATTAAGACGCAG-3′	667 bp	[30]
*ompT*	5′-TCATCCCGGAAGCCTCCCTCACTACTAT-3′ 5′-TAGCGTTTGCTGCACTGGCTTCTGATAC-3′	496 bp	[12]
*hlyF*	5′-GGCCACAGTCGTTTAGGGTGCTTACC-3′ 5′-GGCGGTTTAGGCATTCCGATACTCAG-3′	450 bp	[12]
*iss*	5′-CAGCAACCCGAACCACTTGATG-3′ 5′-AGCATTGCCAGAGCGGCAGAA-3′	323 bp	[30]
*iutA*	5′-GGCTGGACATCATGGGAACTGG-3′ 5′-CGTCGGGAACGGGTAGAATCG-3′	302 bp	[12]

**Table 2 microorganisms-12-01463-t002:** Place of origin of batches according to state and genes detected in each batch.

Brazilian State	Batch Number	Genes Detected
Santa Catarina	1	* ompT * , *iutA*, *iss*, *iroN*, *hlyF*
	4	* ompT * , *iutA*, *iss*, *iroN*, *hlyF*
	5	* ompT * , *iss*, *iroN*, *hlyF*
	6	* ompT * , *iutA*, *iss*, *iroN*, *hlyF*
	7	* iutA * , *iss*, *hlyF*
Rio Grande do Sul	18	* ompT * , *iutA*, *iss*, *iroN*, *hlyF*
	19	* ompT * , *iutA*, *iss*, *iroN*, *hlyF*
	20	* ompT * , *iutA*, *iss*, *iroN*, *hlyF*
	23	* ompT * , *iutA*, *iss*, *hlyF*
	24	* ompT * , *iutA*, *iss*, *hlyF*
	26	* ompT * , *iutA*, *iss*, *iroN*, *hlyF*
	28	* ompT * , *iutA*, *iss*, *iroN*, *hlyF*
	29	* ompT * , *iroN*, *hlyF*
	30	* ompT * , *iutA*, *iss*, *iroN*, *hlyF*
Paraná	31	* ompT * , *iutA*, *iss*, *iroN*, *hlyF*
	33	* ompT * , *iutA*, *iss*, *iroN*, *hlyF*
	35	* ompT * , *iutA*, *iss*, *iroN*, *hlyF*
	38	* ompT * , *iutA*, *iss*, *iroN*, *hlyF*
	39	* ompT * , *iutA*, *iss*, *iroN*, *hlyF*
	40	* iutA * , *iss*, *iroN*, *hly*
	41	* ompT * , *iutA*, *iss*, *iroN*, *hlyF*
	43	* ompT * , *iutA*, *iss*, *hlyF*
	45	* ompT * , *iutA*, *iss*, *iroN*, *hlyF*
	46	* ompT * , *iutA*, *iroN*, *hlyF*
	48	* ompT * , *iutA*, *iss*, *hlyF*
	49	* ompT * , *iutA*, *iss*, *iroN*, *hlyF*
	51	* ompT * , *iutA*, *iss*, *iroN*, *hlyF*
	52	* ompT * , *iutA*, *iss*, *iroN*, *hlyF*
	54	* ompT * , *iutA*, *iss*, *iroN*, *hlyF*
	55	* ompT * , *iutA*, *iss*, *iroN*, *hlyF*
	56	* ompT * , *iutA*, *iss*, *iroN*, *hlyF*
	57	* ompT * , *iutA*, *iss*, *iroN*, *hlyF*
	58	* ompT * , *iutA*, *iss*, *iroN*, *hlyF*
	59	* ompT * , *iutA*, *iss*, *iroN*, *hlyF*
Minas Gerais	61	* ompT * , *iutA*, *iss*, *iroN*, *hlyF*
	65	* ompT * , *iutA*, *iss*, *iroN*, *hlyF*
	66	* ompT * , *iutA*, *iss*, *iroN*, *hlyF*
	67	* ompT * , *iutA*, *iss*, *iroN*, *hlyF*
	69	* ompT * , *iutA*, *iss*, *iroN*
	75	* ompT * , *iutA*, *iss*, *iroN*, *hlyF*
	76	* ompT * , *iutA*, *iss*, *iroN*, *hlyF*
	77	* hlyF *
	78	* hlyF *
	80	* ompT * , *iutA*, *iss*, *iroN*, *hlyF*
São Paulo	62	* ompT * , *iutA*, *iss*, *iroN*, *hlyF*
	63	* ompT * , *iutA*, *iss*, *iroN*, *hlyF*
	64	* ompT * , *iutA*, *iss*, *iroN*, *hlyF*
	68	* ompT * , *iutA*, *iss*, *iroN*
	70	* ompT * , *iutA*, *iss*, *iroN*, *hlyF*
	71	* ompT * , *iutA*, *iss*, *iroN*, *hlyF*
	72	* ompT * , *iutA*, *iss*, *iroN*, *hlyF*
	74	* ompT * , *iutA*, *iss*, *iroN*
	79	* ompT *
Ceará	82	* ompT * , *hlyF*
	83	* ompT * , *iutA*, *iss*, *iroN*, *hlyF*
	87	* ompT * , *iutA*, *iss*
	88	* ompT * , *iutA*, *iss*, *iroN*, *hlyF*
	89	* ompT * , *iutA*, *iss*, *hlyF*
	90	* ompT * , *iutA*, *iss*, *hlyF*
	93	* ompT * , *iutA*, *iss*, *hlyF*
	95	-
	98	* ompT * , *iss*, *iroN*, *hlyF*
	99	* ompT * , *iutA*, *iss*, *iroN*, *hlyF*

**Table 3 microorganisms-12-01463-t003:** Prevalence of virulence genes in *Escherichia coli* isolates.

Gene, Operon, or Region	Description	Prevalence (%)
** Adhesins **		
* eae *	Intimin	1.58
* papA_F11 *	Major pilin subunit F11	3.17
* papA_F19 *	Major pilin subunit F19	1.58
* papA_F20 *	Major pilin subunit F20	11.11
* papC *	Pilus associated with pyelonephritis	23.80
* hra *	Heat-resistant agglutinin	69.84
* iha *	Adherence protein	12.69
* lpfA *	Long polar fimbriae	60.31
* tsh *	Temperature-sensitive hemagglutinin	34.92
** Invasins **		
* ibeA *	Invasion of brain endothelium	1.58
** Serum resistance factors **		
* cvaC *	Structural genes of colicin V operon (Microcin ColV)	30.15
* kpsE *	Capsule polysaccharide export inner-membrane protein	11.11
* kpsMII *	ABC-type polysaccharide/polyol phosphate export system permease; Group 3 capsule	4.76
* kpsMIII_K98 *	Polysialic acid transport protein; Group 2 capsule	1.58
* kpsMII_K1 *	Polysialic acid transport protein; Group 2 capsule	1.58
* kpsMII_K5 *	Polysialic acid transport protein; Group 2 capsule	3.17
* neuC *	Polysialic acid capsule biosynthesis protein	3.17
* traT *	Outer membrane protein complement resistance	82.53
** Iron acquisition systems **		
* chuA *	Heme receptor gene (*E. coli* haem utilization)	41.26
* fyuA *	Siderophore receptor	31.74
* ireA *	Siderophore receptor	20.63
* irp2 *	Iron repressible protein (yersiniabactin synthesis)	31.74
* iucC *	Aerobactin synthetase	53.96
* sitA *	Iron transport protein	61.90
** Toxins **		
* astA *	EAST-1 heat-stable toxin	39.68
* cma *	Structural gene for CoIM activity	23.80
* hlyE *	Avian *E. coli* haemolysin	77.77
* usp *	Uropathogenic-specific protein (bacteriocin)	3.17
* vat *	Vacuolating autotransporter toxin	15.87
** Other virulence factors **		
* air *	Enteroaggregative immunoglobulin repeat protein	12.69
* pic *	Serin protease autotransporter	17.46
* eilA *	Salmonella HilA homolog	12.69
* espA *	Type III secretion system	1.58
* espB *	Secreted protein B	1.58
* espF *	Type III secretion system	1.58
* espJ *	Prophage-encoded type III secretion system effector	1.58
* etpD *	Type II secretion protein	1.58
* etsC *	Putative type I secretion outer membrane protein	73.01
* capU *	Hexosyltransferase homolog	1.58
* cba *	Colicin B	3.17
* cea *	Colicin E1	17.46
* celb *	Endonuclease colicin E2	4.76
* cia *	Colicin ia	30.15
* cib *	Colicin ib	17.46
* cif *	Type III secreted effector	1.58
* gad *	Glutamate decarboxylase	100
* mchB *	Microcin H47 part of colicin H	1.58
* mchC *	MchC protein	1.58
* mchF *	ABC transporter protein MchF	23.80
* mcmA *	Microcin M part of colicin H	1.58
* nleA *	Non-LEE encoded effector A	1.58
* nleB *	Non-LEE encoded effector B	1.58
* tccP *	Tir-cytoskeleton coupling protein	1.58
* terC *	Tellurium ion resistance protein	100
* tir *	Translocated intimin receptor protein	1.58
* yfcV *	Fimbrial protein	6.34

**Table 4 microorganisms-12-01463-t004:** Prevalence of antimicrobial resistance genes (ARGs) in *Escherichia coli* isolates.

Antimicrobial Class	Antimicrobial Resistance Gene	Prevalence (%)
** Aminoglycosides **	* aac(3)-IId *	4.76
	* aac(3)-IVa *	3.17
	* aac(3)-VIa *	36.5
	* aadA12 *	1.58
	* aadA2 *	20.63
	*ant(2*″*)-Ia*	7.93
	*ant(3*″*)-Ia*	50.79
	* aph(3 * ′ * )-Ia *	20.63
	*aph(3*″*)-Ib*	31.74
	* aph(4)-Ia *	3.17
	* aph(6)-Id *	31.74
** Beta-lactams **	* blaCMY-2 *	9.52
	* blaCTX-M-1 *	3.17
	* blaCTX-M-15 *	1.58
	* blaCTX-M-164 *	1.58
	* blaCTX-M-2 *	17.46
	* blaCTX-M-55 *	9.52
	* blaCTX-M-8 *	12.69
	* blaSHV-12 *	1.58
	* blaTEM-106 *	1.58
	* blaTEM-141 *	6.34
	* blaTEM-1A *	7.93
	* blaTEM-1B *	14.28
** Trimethoprim **	* dfrA1 *	7.93
	* dfrA12 *	4.76
	* dfrA14 *	4.76
	* dfrA15 *	4.76
	* dfrA7 *	3.17
	* dfrA8 *	3.17
** Phenicoles **	* catA1 *	3.17
	* cmlA1 *	9.52
	* floR *	19.04
** Lincosamides **	* lnu(A) *	4.76
	* lnu(F) *	7.93
** Colistin **	* mcr-1.5 *	1.58
	* mcr-9 *	1.58
** Macrolides **	* mph(A) *	1.58
	* mph(B) *	1.58
** Quinolones **	* qnrA1 *	1.58
	* qnrB19 *	11.11
	* qnrS1 *	6.34
** Sulfonamides **	* sul1 *	52.38
	* sul2 *	60.31
	* sul3 *	7.93
** Tetracyclines **	* tet(A) *	31.74
	* tet(B) *	20.63
	* tet(D) *	1.58
** Others **	* fosA *	6.25
	* fosA3 *	1.5625
	* qacE *	50.79
	* sitABCD *	57.14
	*formA (Genbank Acc, No, X73835)*	100

## Data Availability

The genome sequencing and assembly data were deposited in the NCBI database with a Bioproject accession number (PRJNA917297).

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
