# Peer review of "Isolation and Characterization of Escherichia coli from Brazilian Broilers"

_microorganisms, 2024, doi:10.3390/microorganisms12071463_

Round 1
Reviewer 1 Report
Comments and Suggestions for Authors
The manuscript is interesting above all because it can be used as background to carry out APEC monitoring and establish control and/or prevention measures. However, the authors must consider some points.
Introduction
L.34. Include reference.
L.57. Check the writing.
L.63-64. Check the writing.
Materials and methods
L.102.In the abstract (L.19) it is shown that the samples were taken between August and November and here the authors place that between August and December. What is the correct period?
L.105.How many birds did each batch contain?
L.112-116. Check the writing of this paragraph.
L.131. Include the meaning of "LB".
L.152. Include supplier, city and country.
L.181. Isn't it better to use a vernier for greater precision?
Results
L.243-251. Considering this background, are the authors underestimating results or not?
L.254. Why did the authors decide to only analyze 63 of 65 isolates?
L.260. "four" must be in number "4".
L.261.It is important to indicate that in batch 95 there was no detection of genes.
L.270. A figure title cannot be very large. There is information in the title that should be described in a paragraph. It is not necessary to include figure 1 or 2 or 3 or 4.
L.292. of what???
L.309. 100%???
L.327. Check that the resistance genes are found in both the text and the table.
L.329-330. Improve writing.
L.335. aadA5 is not shown in the table, is this correct?
L.340. blaSHV-187 is not shown in the table, is this correct?
Discussion
L.365. Why at least 3 and not 1, 2 or 4?
L.400. of the...

Author Response
Journal Microorganisms (ISSN 2076-2607)
Manuscript ID: microorganisms-3081689
Title: Isolation and characterization of Escherichia coli from Brazilian broilers
By Pilati et al.
Response to reviews.
We appreciate the review. The text has been extensively revised for grammar and content details.
Below we highlight the main revisions included.
#Response to Reviewer 1 Comments
Dear Reviewer,
Our team is grateful for all the suggestions.
The text has been extensively revised for grammar and content details. Below we highlight the main revisions included in the text (view MS in yellow color), as well as the manuscript with the requested changes.
Responses:
Introduction:
L.34. Include reference.
Response: Thank you for the observation. The following reference was added: ABPA (Brazilian Animal Protein Association) Annual Report 2024; 2024. number “[1]”.
L.57. Check the writing.
Response: Thank you for your feedback. The text has been revised and improved.
“When secondary, APEC infections can be associated with respiratory viruses such as Newcastle disease, Infectious Bronchitis Virus, and avian Metapneumovirus, or with bacteria such as Mycoplasma gallisepticum and Mycoplasma synoviae [5,8,9].”
L.63-64. Check the writing.
Response: Thank you for your feedback. The text has been revised and improved.
“APEC strains utilize different virulence and pathogenicity factors to establish an infection and cause disease in birds. The main virulence factors include proteins that facilitate adhesion and invasion, elements involved in toxin production, secretion systems, and mechanisms for antibiotic resistance [5,11].”
L.102.In the abstract (L.19) it is shown that the samples were taken between August and November and here the authors place that between August and December. What is the correct period?
Response: Thank you for the observation. Sample collections took place from August to November. The methodology information (line 102) has been corrected.
L.105. How many birds did each batch contain?
Response: Thank you for the observation. Each batch contains approximately 10 thousand birds.
L.112-116. Check the writing of this paragraph.
Response: Thank you for the observation. The text has been revised and improved.
“The femurs were chosen for the isolation of E. coli because they were collected intact, which reduced the chances of bone marrow contamination.”
L.131. Include the meaning of "LB".
Response: Thank you for the observation. The meaning of LB has been added to the text in line 132.
“Isolates were preserved in a repository and biobank in Luria Bertani (LB) broth (KASVI, Spain) with glycerol (5:1), and maintained at -80 °C for subsequent genomic DNA extraction.”
L.152. Include supplier, city, and country.
Response: Thank you for the correction. Supplier, city, and country were added.
“(QIAGEN, Inc., Valencia, EUA)”.
L.181. Isn't it better to use a vernier for greater precision?
Response: Thank you for the observation. We used the wrong term when writing "millimetric ruler." The term was corrected in the text.
Results:
L.243-251. Considering this background, are the authors underestimating results or not?
Response: Thank you for this consideration. To clarify this point, the information was included in the Materials and Methods section. The study selected birds with respiratory clinical signs, but with an unknown etiology, meaning that not all of them were sick due to E. coli infection.
“From the batches analyzed, 61% had a history of respiratory problems, septicemia, and/or mortality, with unknown etiology. ”
L.254. Why did the authors decide to only analyze 63 of 65 isolates?
Response: Thank you for the observation. A typographical error occurred, and the number has been corrected.
“A total of 63 isolates characteristic of E. coli were obtained from the femurs, and these isolates were subjected to whole-genome sequencing.”
L.260. "four" must be in number "4".
Response: Thank you for the correction. The number has been corrected.
L.261.It is important to indicate that in batch 95 there was no detection of genes.
Response: We appreciate the observation. The following sentence has been added to the text:
“and in one batch, none of the five genes were detected”.
L.270. A figure title cannot be very large. There is information in the title that should be described in a paragraph. It is not necessary to include figure 1 or 2 3 or 4.
Response: We appreciate the observation. The text used as the image title was improved and expanded into paragraphs.
“The obtained diameters were compared with the cutoff points established by the CLSI [31,32].
The overall resistance rates found were 66.67% for Ampicillin, 7.94% for Azithromycin, 44.44% for Ceftriaxone, 44.44% for Ceftiofur, 39.68% for Enrofloxacin, 30.16% for Gentamicin, 19.95% for Lincomycin/Spectinomycin, 69.84% for Nalidixic Acid, 7.94% for Nitrofurantoin, 14.29% for Norfloxacin, and 42.86% for Sulfamethoxazole/Trimethoprim.
In the South region (Figure 3), the resistance rates found for E. coli isolates were 64.71% for Ampicillin, 2.94% for Azithromycin, 47.06% for Ceftriaxone, 47.06% for Ceftiofur, 32.35% for Enrofloxacin, 23.53% for Gentamicin, 14.71% for Lincomycin/Spectinomycin, 61.76% for Nalidixic Acid, 2.94% for Nitrofurantoin, 8.82% for Norfloxacin and 26.47% for Sulfazotrim.
In the Southeast region, the results found for E. coli isolates indicate resistance to Ampicillin of 57.89%, 47.37% for Ceftriaxone, 47.37% for Ceftiofur, 44.11% for Enrofloxacin, 26.32% for Gentamicin, 31.58% for Lincomycin/Spectinomycin, Nalidixic Acid of 68.42%, 5.26% for Nitrofurantoin, 5.26% for Norfloxacin and 63.16% for Sulfazotrim, all isolates were sensitive to Azithromycin.
In the Northeast region (Figure 4), the resistance indices found for E. coli isolates indicate resistance to Ampicillin of 70%, 30% for Azithromycin, 30% for Ceftiofur, 30% to Ceftriaxone, 60% for Enrofloxacin, 40% for Gentamicin, 10% for Lincomycin/Spectinomycin, Nalidixic Acid of 80%, 20% for Nitrofurantoin, 40% for Norfloxacin and 40% for Sulfazotrim.
Figure 1 represents the general resistance rates and the rates for each region.
Figure 1. Resistances presented by APEC isolates against the tested antimicrobials, where: Ampicillin (AMP), Azithromycin (AZI), Ceftiofur (CFT), Ceftriaxone (CRO), Enrofloxacin (ENRO), Gentamicin (GEN), Lincomycin/Spectinomycin (LIN+SPT), Nalidixic Acid (NAL), Nitrofurantoin (NIT), Norfloxacin (NOR), and Sulfazotrim (SUT). (a) Resistances presented by APEC isolates against the tested antimicrobials in Brazil; (b) Resistances presented by the APEC isolates from the South region; (c) Resistances presented by APEC isolates from the Southeast region; (d) Resistances presented by APEC isolates from the Northeast region.”
L.292. of what???
Response: Thank you for the correction. The term “of” was placed incorrectly.
L.309. 100%???
Response: We appreciate the observation. All isolates, 100% carried at least one gene encoding toxins. The percentage has been added to the sentence.
“All isolates (100%) carried at least one gene encoding toxins (astA, cma, cvaC, hlyE, hlyF, usp, vat).”
L.327. Check that the resistance genes are found in both the text and the table.
Response: Thank you for the correction. The sentences have been corrected to align with the table.
L.329-330. Improve writing.
Response: Thank you for the observation. The text has been revised and improved.
“Out of the analyzed isolates, all of them showed at least one antimicrobial resistance gene (ARG), and all of them (100%) contained the formA gene, a formaldehyde resistant gene.”
L.335. aadA5 is not shown in the table, is this correct?
L.340. blaSHV-187 is not shown in the table, is this correct?
Response: Thank you for the correction. The genes aadA5 and blaSHV-187 were absent in the isolates. The sentences have been corrected to align with the table.
Discussion:
L.365. Why at least 3 and not 1, 2 or 4?
Response: We appreciate the observation. This is the minimum number determined by the cited reference (Reference [12]) for a strain to be considered pathogenic to birds.
L.400. of the…
Response: Thank you for the observation. The sentence have been corrected.
Please see the attachment.

Reviewer 2 Report
Comments and Suggestions for Authors
Please see the attachment.

/
